# UniDense: Unleashing Diffusion Models with Meta-Routers for Universal Few-Shot Dense Prediction

## ABSTRACT

Universal few-shot dense prediction requires a versatile model capable of learning any dense prediction task from limited labeled images, which necessitates the model to possess efficient adaptation abilities. Prevailing few-shot learning methods rely on efficient fine-tuning of model weights for few-shot adaptation, which carries the risk of disrupting the pre-trained knowledge and lacks the capability to extract task-specific knowledge contained in the pre-trained model. To overcome these limitations, our paper approaches universal few-shot dense prediction from a novel perspective. Unlike conventional fine-tuning techniques that directly use all parameters of the model and modify a specific set of weights for few-shot adaptation, our method focuses on selecting the task-relevant computation pathways of the pre-trained model while keeping the model weights frozen. Building upon this idea, we introduce a novel framework UniDense for universal few-shot dense prediction. First, we construct a versatile MoE architecture for dense prediction based on the Stable Diffusion model. We then utilize episodes-based meta-learning to train a set of routers for this MoE model, called Meta-Routers, which act as hyper-networks responsible for selecting computation blocks relevant to each task. We demonstrate that fine-tuning these meta-routers for novel tasks enables efficient adaptation of the entire model. Moreover, for each few-shot task, we leverage support samples to extract a task embedding, which serves as a conditioning factor for meta-routers. This strategy allows meta-routers to dynamically adapt themselves for different few-shot task, leading to improved adaptation performance. Experiments on a challenging variant of Taskonomy dataset with 10 dense prediction tasks demonstrate the superiority of our approach.

## CCS CONCEPTS

• **Computing methodologies → Computer vision**; **Machine learning**.

## KEYWORDS

few-shot learning, dense prediction, diffusion model, mixture of experts, multi-task pre-training

## 1 INTRODUCTION

Dense prediction tasks, such as semantic segmentation, depth estimation and edge detection, hold significant importance in the field

of computer vision. Their objective is to learn a mapping from input images to pixel-wise annotated labels. Given the high cost of pixel-wise labeling for supervised methods, there is a strong demand for developing a few-shot learner which can flexibly and efficiently learn any dense task from a few labeled images. This specific task is referred to as universal few-shot dense prediction [18], which has been recently proposed. The ultimate objective of few-shot learning is to emulate the human brain's capacity to learn arbitrary new tasks with minimal samples. The task of universal few-shot dense prediction serves as a testing ground for evaluating such learning approaches in the domain of dense prediction tasks.

A crucial requirement for universal few-shot dense prediction is the ability of the model to adapt flexibly to diverse unseen tasks while being efficient enough to avoid over-fitting [18]. Several few-shot learning methods, including metric learning and in-context learning, have emerged to address this challenge by enabling rapid adaptation to novel few-shot tasks without the need for additional fine-tuning [10, 36, 37, 40]. However, numerous experiments consistently demonstrate that models with fine-tuning still achieve the highest performance [15, 20, 21, 24]. The naive approach involves fine-tuning the entire model, as depicted in Figure 1a. Nevertheless, the limited samples in the few-shot setting is insufficient for full-model fine-tuning, resulting in overfitting. To address this issue, current fine-tuning methods often resort to partial fine-tuning. A common approach is to fine-tune the last few layers [1, 38, 42], as shown in Figure 1b. Some methods also fine-tune the biases [2, 18, 44], as depicted in Figure 1c. However, all these methods achieve few-shot adaptation by modifying the model weights, which may disrupt the well-learned knowledge from the pre-training stage. Moreover, recent experiments have demonstrated that not all features of a pre-trained model are beneficial for novel few-shot tasks and selecting task-specific features may lead to improved performance [4, 22]. However, prevalent fine-tuning methods simply use all parameters of the model and lack the ability to extract pre-existing task-specific knowledge embedded within the pre-trained model.

To overcome these limitations, our paper approaches universal few-shot dense prediction from a novel perspective. Unlike conventional fine-tuning techniques that directly use all parameters of the model and modify a specific set of weights to achieve few-shot adaptation, our method focuses on selecting the task-specific computation pathways of the pre-trained model while keeping the model weights frozen, as shown in Figure 1d. Building upon this concept, we introduce a novel framework called UniDense for universal few-shot dense prediction. The basic idea of this framework is to construct a MoE (Mixture of Experts) backbone and achieve few-shot adaptation by fine-tuning its routers, which act as hyper-networks responsible for selecting task-specific computation blocks. Specifically, our UniDense framework follows a three-step pipeline: (1) We transform the UNet-like denoising autoencoder from Stable Diffusion (SD) [29] into a MoE model. This design allows us to leverage SD pre-trained knowledge, which has been

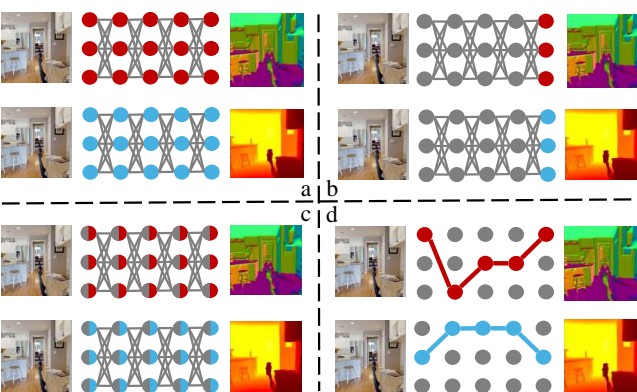

Figure 1: Comparison of different few-shot adaptation strategies: Full-model fine-tuning (a) to adjust all model weights for different tasks; Prediction head fine-tuning (b) and bias tuning (c) to adjust partial model weights; Our adaptation method (d) to select task-relevant computation pathways.

empirically proven to be useful for a wide range of dense prediction tasks [1, 41, 48]. We then conduct multi-task training on this MoE model using datasets from various base training tasks (disjoint from testing tasks). After this, we assume that the model possesses the necessary general knowledge to tackle arbitrary dense prediction tasks. (2) We utilize episodes-based meta-learning to train a set of routers for this MoE model, called Meta-Routers. Each episode is designed to simulate the few-shot setting of testing phase, enabling the meta-routers to learn how to efficiently fine-tune on arbitrary unseen dense prediction tasks. (3) For novel few-shot tasks, we fine-tune these meta-routers to achieve efficient adaptation of the entire model by choosing task-specific computation pathways.

Moreover, for each few-shot task, we leverage support samples to extract a task embedding, which serves as a conditioning factor for the meta-routers. This strategy allows meta-routers to dynamically adapt themselves for different few-shot task, instead of rely solely on fine-tuning a fixed set of initialization weights, thus increasing their flexibility and leading to improved adaptation performance.

We follow the standard evaluation setup of universal few-shot dense prediction [18] to validate our method. A variant of the Taskonomy dataset [45], consisting of ten tasks, is used for evaluation. The dataset is partitioned into a 5-fold split, where two tasks are selected for few-shot evaluation in each split, while the remaining eight tasks are used for training. This setup effectively simulates the few-shot learning scenario for unseen dense prediction tasks. Experimental results across all splits consistently demonstrate the superior performance of our proposed approach.

Our paper makes the following key contributions: (1) We introduce UniDense, a novel framework for universal few-shot dense prediction, featuring a reliable and versatile MoE architecture based on the Stable Diffusion model, along with the technique of router fine-tuning to achieve efficient adaptation by selecting task-specific computation pathways instead of modifying model weights. (2) We propose meta-routers which are trained with episodes-based meta-learning to learn how to rapidly adapt. Moreover, a task embedding directly generated from the few-shot samples is used to dynamically adjust meta-routers, resulting in highly efficient model adaptation. (3) Experimental results on a variant of the Taskonomy

dataset across all dataset splits and various shot numbers consistently demonstrate the superior performance of our approach.

## 2 RELATED WORK

**Universal few-shot dense prediction.** Existing few-shot learning methods in computer vision are typically designed for specific tasks like classification and semantic segmentation [17, 23, 39]. However, these methods often rely on task-specific prior knowledge and assumptions in their model architecture and training procedures, making them less suitable for generalizing to arbitrary dense prediction tasks. VTM [18] is the pioneering work that addresses the challenge of few-shot learning for arbitrary dense prediction tasks in a universal manner, which utilizes non-parametric matching on tokenized image and label embeddings. However, VTM's computational demands and data inefficiency arise from the matching operation between all tokens of the query image and the support images. Furthermore, the bias tuning technique used for few-shot adaptation in VTM lacks the ability to excavate the task-specific knowledge embedded in the pre-trained model. In contrast, our UniDense framework offers rapid inference regardless of the number of support images, and the router fine-tuning technique allows our method to extract task-specific knowledge from the pre-trained model by selecting task-relevant computation pathways.

**Few-shot fine-tuning.** Conventional fine-tuning strategies for few-shot learning achieve efficient adaptation by selectively fine-tuning specific parts of the model [18, 19, 35, 38]. Another set of methods involve inserting adapter modules into the frozen pre-trained model and fine-tuning these adapters for few-shot adaptation [12, 46]. However, all these methods directly use all parameters of the model and overlook the extraction of task-specific knowledge hidden in the well-learned pre-trained model. In contrast, our approach achieves few-shot adaptation by selectively choosing computation pathways that excavate the task-specific knowledge. The closest relevant studies to our method involve training a masking module to select relevant features for novel few-shot tasks [4, 49]. However, these methods are not tailored for the model with multi-task pre-training, which limits their ability for universal few-shot dense prediction. In contrast, our MoE design allows us to fully exploit the knowledge from multiple base training tasks.

**Unleashing diffusion models for visual perception.** While diffusion models are primarily trained using generative loss, their features have shown impressive performance in specific visual perception tasks that demand a comprehensive understanding of pixel-level fine-grained information [1, 41, 48], such as semantic segmentation and depth estimation. However, existing works either focus on leveraging diffusion models for a specific task [1, 41] or address multiple visual perception tasks without considering the few-shot setting [48]. Currently, there is no effective and elegant method to fully harness the pre-trained knowledge of diffusion models and adapt them to few-shot visual perception tasks in a universal manner. To the best of our knowledge, we are the first to make an attempt to bridge this gap.

## 3 PROBLEM DEFINITION

The problem of universal few-shot dense prediction [18] aims to build a universal few-shot learner $\mathcal{F}$ that, for any dense prediction

task $\mathcal{T} : \mathbb{R}^{H \times W \times 3} \rightarrow \mathbb{R}^{H \times W \times C_{\mathcal{T}}}$, where $C_{\mathcal{T}} \in \mathbb{N}$, can produce a prediction $\hat{Y}_Q$ for an unseen (query) image $X_Q$ given only a few labeled (support) examples $\mathcal{S}_{\mathcal{T}}$:

$$\hat{Y}_Q = \mathcal{F}(X_Q; \mathcal{S}_{\mathcal{T}}), \quad \mathcal{S}_{\mathcal{T}} = \{(X^i, Y^i)\}_{i \leq N_S}, \quad (1)$$

where $N_S$ denotes the number of labeled images for each few-shot dense prediction task, namely the shot number.

To capture the general knowledge for the domain of dense prediction, several base dense prediction tasks $\{\mathcal{T}_B^i\}_{i \leq N_B}$ are used for the multi-task pre-training of the few-shot learner $\mathcal{F}$, where each base task does not have a limitation on the number of examples. Here, $N_B$ represents the number of base tasks and the testing task is not included among the base tasks.

## 4 METHODOLOGY

In this section, we present the details of our proposed UniDense framework. We begin by introducing the foundational concepts of MoE (Mixture of Experts) and Stable Diffusion model, upon which UniDense is built. Next, we delve into the architectural design details of our method. Subsequently, we proceed to introduce the three-stage pipeline of our UniDense framework.

### 4.1 Preliminaries

**Mixture of Experts.** A MoE layer typically contains a set of expert networks $E^1, E^2, ..., E^{N_E}$ along with a routing network (router) $G$. The output of a MoE layer is the weighted sum of the output from every expert. The router $G$ calculates the weight $G^k(x)$ for each expert given the input $x$. Formally, the output of a MoE layer is

$$y = \sum_{k=1}^{N_E} G^k(x) E^k(x). \quad (2)$$

The router $G$ is a noisy top-K routing network [34] with parameters $W_G$ and $W_N$. It models $P(E^k|x)$ as the probability of using expert $E^k$ and selects $N_K$ experts with the highest $P(E^k|x)$ to contribute to the final output. The whole process is shown as follows:

$$G(x) = Top(Softmax(xW_G + \mathcal{N}(0, 1)Softplus(xW_N)), N_K), \quad (3)$$

where $Top(\cdot, N_K)$ sets all elements in the vector to zero except the elements with the largest $N_K$ values. $\mathcal{N}(0, 1)$ represents sampling from a normal distribution. Softplus is the smooth approximation to the ReLU function:

$$Softplus(x) = log(1 + exp(x)). \quad (4)$$

**Stable Diffusion.** The Stable Diffusion model [29] is a text-to-image model trained on the large-scale image-text dataset LAION-5B [31]. This model has demonstrated impressive performance in generating images controlled by language descriptions. Initially, they train a VQGAN model [7], which comprises an encoder $\mathcal{E}$ and a decoder $\mathcal{D}$, enabling conversion between the pixel space and the latent space. Subsequently, a diffusion model is trained on this latent space. The noise predictor network $\epsilon_\theta$ employed in the latent diffusion model is implemented as a UNet [30] and incorporates rich pre-trained knowledge, which has empirically shown its effectiveness in dense prediction tasks [1, 41, 48]. Hence, we opt to adopt the UNet architecture as the foundation for our model's backbone. Our approach does not utilize the diffusion process and language

interface of the Stable Diffusion model. Thus, we leave the detail descriptions of the Stable Diffusion model in the appendix.

### 4.2 Architecture

Our model architecture consists of a MoE backbone based on the UNet-like denoising autoencoder from the Stable Diffusion model, along with a prediction head that produces predictions with varying dimensions for different tasks, as shown in Figure 2.

**Backbone.** To leverage the pre-trained knowledge of the Stable Diffusion model, our backbone is constructed based on the UNet-like denoising autoencoder $\epsilon_\theta$ from Stable Diffusion. Specifically, given an input image $X \in \mathbb{R}^{H \times W \times 3}$, we firstly employ the encoder $\mathcal{E}$ of the VQGAN to encode the input image into the latent space:

$$Z = \mathcal{E}(X), \ Z \in \mathbb{R}^{\frac{H}{8} \times \frac{W}{8} \times 4}. \quad (5)$$

We feed the latent feature map $Z$ to $\epsilon_\theta$ and then a set of hierarchical feature maps are extracted from the last layer of each output block in different resolutions:

$$\{F^i\}_{i \leq 4} = \epsilon_\theta(Z), \quad (6)$$

where $F^i \in \mathbb{R}^{\frac{H}{2^{i+2}} \times \frac{W}{2^{i+2}} \times C^i}$, with $i = 1, 2, 3, 4$. For instance, if the input image size is 256×256, the output feature maps of $\epsilon_\theta$ will have side lengths of 32, 16, 8, 4. Note that we only use the self-attention layers in $\epsilon_\theta$ since we do not utilize the language interface of the Stable Diffusion model. Additionally, no extra noise is added to the input image, which is equivalent to using a timestep of 0 in the diffusion process.

To enable efficient adaptation by selecting task-specific computation pathways for novel few-shot tasks, we propose to fine-tune the routers of a MoE backbone. Here, we show how to construct the MoE backbone based on the Stable Diffusion model. Inspired by Mod-Squad [3], which transforms the vision transformer into a MoE model, we convert the multi-head attention layers and the feed-forward layers in the transformer blocks of the Stable Diffusion model's denoising autoencoder $\epsilon_\theta$ into MoE attention layers and MoE MLP layers.

We follow MoA [47] to construct the MoE attention layers. Given a query token $q^i \in \mathbb{R}^{1 \times C_M}$ from the query sequence $Q$, the objective of a MoE attention layer is to generate a new token $y^i$ for $q^i$, with $N_E^a$ attention experts $\{E_A^k\}_{k=1}^{N_E^a}$ and the router $G_A$:

$$y^i = \sum_{k=1}^{N_E^a} G_A^k(q^i) \cdot E_A^k(q^i), \quad (7)$$

$$E_A^k(q^i) = (\alpha^{i,k} V W_V) W_O^k, \quad (8)$$

$$\alpha^{i,k} = Softmax(\frac{q^i W_Q^k (K W_K)^T}{\sqrt{C_H}}), \quad (9)$$

where $C_H$ is the head dimension. $W_K, W_V \in \mathbb{R}^{C_M \times C_H}$ are shared across attention experts to reduce computational complexity, while $W_Q^k \in \mathbb{R}^{C_M \times C_H}$ and $W_O^k \in \mathbb{R}^{C_H \times C_M}$ are specific to each expert.

As for a MoE MLP layer, we convert the GEGLU feed-forward layer [33] adopted in $\epsilon_\theta$ into a MoE model:

$$y = \sum_{k=1}^{N_E^f} G_F^k(x) \cdot E_F^k(x), \quad (10)$$

**A. Multi-Task Pre-Training for MoE Backbone with Task-Specific Routers**

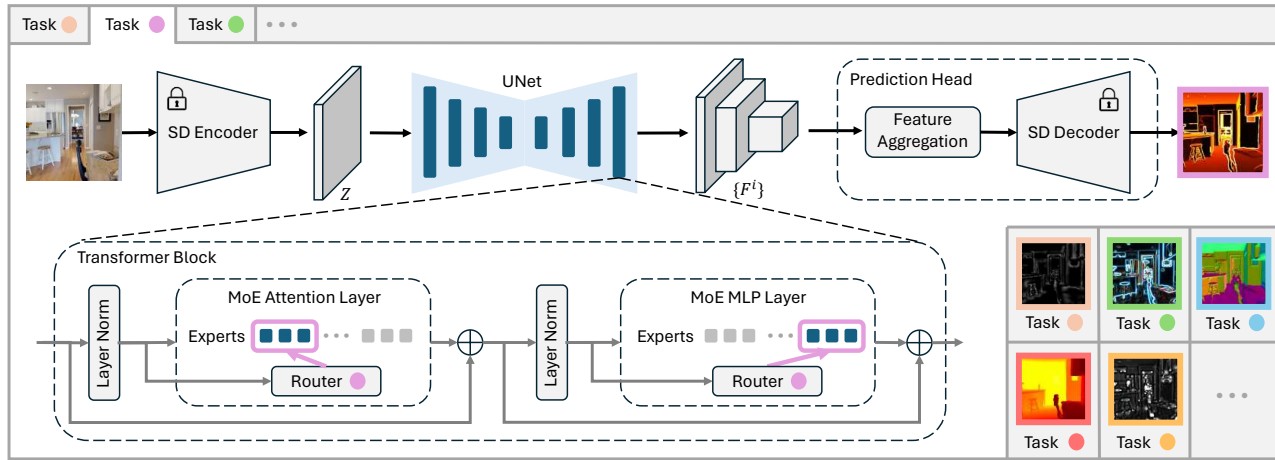

**B. Episodes-Based Meta-Training for Meta-Routers**

**C. Fine-Tuning Meta-Routers for Novel Few-Shot Tasks**

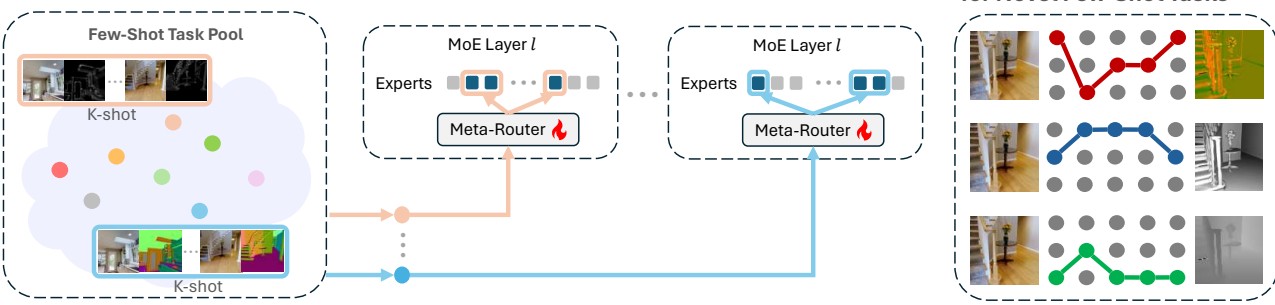

**Figure 2: Illustration of our proposed UniDense framework, which consists of three stages (A~C). 'SD Encoder' and 'SD Decoder' represent the VQGAN encoder and decoder from the Stable Diffusion (SD) model. The task-specific convolution layer in SD decoder is omitted for clarity. The 'UNet' is a MoE module transformed from the UNet-like denoising autoencoder of SD model. The 'lock' icon represents freezing parameters and the 'flame' icon represents parameter tuning.**

$$E_F^k(x) = (\Phi(x\hat{W}_G^k) \odot x\hat{W}_I^k)\hat{W}_O^k, \tag{11}$$

where $\Phi$ represents the GELU activation function [13]. $\Phi(x\hat{W}_G)$ act as gating values. $\odot$ represents element-wise product.

The weights of attention experts $\{E_A^k\}$ and MLP experts $\{E_F^k\}$ are initialized using the weights from the multi-head attention layers and the feed-forward layers in $\epsilon_\theta$. For the specific details of this initialization process, please refer to the appendix.

**Prediction head.** Recent research [1] has shown that different levels of feature maps from $\epsilon_\theta$ contain varying levels of semantic information. To aggregate these levels of information, we introduce a feature aggregation module $\mathcal{U}$, which consists of several convolutional layers and upsampling layers. From the lowest scale to the highest scale of $\{F^i\}_{i\le4}$, the feature aggregation module upsamples the current level of feature map and concatenates it with the next level of feature map, followed with convolutions. This procedure generates a feature map with the shape $\mathbb{R}^{\frac{H}{8} \times \frac{W}{8} \times C_\mathcal{U}}$. To generate the final predictions with a high resolution of $H \times W$, an upsampling procedure is required. Instead of using the naive bilinear interpolation, we utilize the decoder $\mathcal{D}$ from the Stable Diffusion model as a parametric upsampling procedure. Since the output dimension $C_\mathcal{T}$ may vary for different tasks, we replace the last convolution layer in $\mathcal{D}$, which originally produces an output with the shape $\mathbb{R}^{H \times W \times 3}$,

with task-specific convolution layers. Therefore, the prediction head can be represented as:

$$\hat{Y} = \mathcal{D}_\mathcal{T}(\mathcal{V}(\mathcal{U}(\{F^i\}_{i\le4}))), \tag{12}$$

where $\mathcal{V}$ denotes a $1 \times 1$ convolution layer that transforms the feature dimension from $\mathbb{R}^{\frac{H}{8} \times \frac{W}{8} \times C_\mathcal{U}}$ to $\mathbb{R}^{\frac{H}{8} \times \frac{W}{8} \times 4}$. $\mathcal{D}_\mathcal{T}$ denotes the Stable Diffusion decoder with the task-specific convolution layer, which produces the final prediction $\hat{Y}$ with the shape $\mathbb{R}^{H \times W \times C_\mathcal{T}}$.

### 4.3 Pipeline

The overall pipeline of our method is illustrated in Figure 2, which can be divided into three stages: multi-task pre-training of the backbone, meta-training for the meta-routers, and fine-tuning of meta-routers for efficient adaptation to novel few-shot tasks.

**Multi-task pre-training.** To capture the general knowledge for the domain of dense prediction, we perform multi-task pre-training for the backbone on a set of base dense prediction tasks $\{\mathcal{T}_B^i\}_{i\le N_B}$. There are two options to achieve multi-task pre-training for a MoE model. The first is to train a specific router for each task, while the other is to train a single router for all tasks conditioned on learnable task embeddings. Experiments have demonstrated that the former is more effective than the latter [8]. Therefore, we assign each dense

prediction task a task-specific router for multi-task pre-training. Note that the Stable Diffusion encoder $\mathcal{E}$ and decoder $\mathcal{D}$ used in our model are fixed during the multi-task pre-training to preserve the pre-trained knowledge from the Stable Diffusion model.

In the single-task setting, previous MoE methods [9, 28, 34] often use a load-balancing loss to promote similar utilization frequencies of experts across different batches. However, in the multi-task setting, this approach may force experts on conflicting tasks with learning gradients counteracting each other [3]. Therefore, we follow Mod-Squad [3] to utilize a mutual information loss to encourage a sparse but strong dependence between experts and tasks:

$$\mathcal{L} = \sum_{i=1}^{N_B} w_{\mathcal{T}_B^i} \mathcal{L}_{\mathcal{T}_B^i} - w_{MI} \sum_{\forall MoE \; layer \; l} I(T; E_l), \tag{13}$$

where $T = \{\mathcal{T}_B^i\}_{i \leq N_B}$, $E_l$ represents all experts in the MoE layer $l$, and $I(T; E_l)$ is the mutual information measurement between tasks $T$ and experts in $E_l$. An auto-balancing weight $w_{\mathcal{T}_B^i}$ is learned for each task-specific loss $\mathcal{L}_{\mathcal{T}_B^i}$. We use cross-entropy loss for semantic segmentation task and L1 loss for the others in our experiments. The hyper-parameter $w_{MI}$ is set as 0.001. Here, we omit the computation details for clarity, which can be found in [3].

**Meta-training for Meta-Routers.** To enable efficient adaptation by selecting task-specific computation pathways, we propose to fine-tune the routers of the MoE backbone. However, it raises questions about which router to fine-tune and how to ensure its effectiveness. A naive approach is to randomly initialize a set of routers. However, due to the limited number of samples in the few-shot testing stage, such randomly initialized routers cannot ensure the adaptation efficiency for diverse novel tasks. To overcome this, we introduce Meta-Routers, which are optimized to efficiently adapt to any novel few-shot task. To obtain meta-routers, we freeze the well pre-trained backbone and employ episodes-based meta-learning to train an additional set of routers $\{G_M^l : \forall MoE \; layer \; l\}$. In each episode, we simulate the few-shot setting during the testing phase. First, we randomly select a base task $\hat{\mathcal{T}}$ and sample $N_S$ examples as the support set $\mathcal{S}_{\hat{\mathcal{T}}}$. Then, we sample another $N_Q$ examples as the query set $Q_{\hat{\mathcal{T}}}$. The objective of the episodes-based meta-learning is to make the model obtain optimal performance on the query set $Q_{\hat{\mathcal{T}}}$ after the few-shot fine-tuning based on the support set $\mathcal{S}_{\hat{\mathcal{T}}}$:

$$\mathcal{L}_M = \frac{1}{|Q_{\hat{\mathcal{T}}}|} \sum_{(X_Q, Y_Q) \in Q_{\hat{\mathcal{T}}}} \mathcal{L}_{\hat{\mathcal{T}}}(Y_Q, \hat{\mathcal{F}}(X_Q; \mathcal{S}_{\hat{\mathcal{T}}})), \tag{14}$$

where $\hat{\mathcal{F}}(X_Q; \mathcal{S}_{\hat{\mathcal{T}}})$ represents the model's prediction for the query image $X_Q$ after fine-tuning $N_{FT}$ steps on the support set $\mathcal{S}_{\hat{\mathcal{T}}}$, and $\mathcal{L}_{\hat{\mathcal{T}}}$ is the task-specific loss. Note that the fine-tuning procedure of $\hat{\mathcal{F}}(X_Q; \mathcal{S}_{\hat{\mathcal{T}}})$ only involves the meta-routers and the task-specific convolution layer in the prediction head. Calculating the gradient of $\mathcal{L}_M$ is a bi-level optimization problem, which involves computing the Jacobian matrix of the fine-tuning procedure $\hat{\mathcal{F}}(X_Q; \mathcal{S}_{\hat{\mathcal{T}}})$, leading to extra computation cost [11]. In this regard, we opt for FOMAML (First-Order Model-Agnostic Meta-Learning) [11, 25] to strike a balance between speed and effectiveness, which can achieve nearly the same performance with MAML [11] while significantly reducing GPU memory requirements and training time, particularly when the number of fine-tuning steps is large [26].

Moreover, we enhance the meta-routers' capabilities by incorporating the task embedding extracted from the support samples. This transforms each meta-router $G_M^l$ into a task-conditioned router:

$$\hat{G}_M^l(x, \mathcal{S}_{\mathcal{T}}) = G_M^l(x + \xi(\mathcal{S}_{\mathcal{T}})), \tag{15}$$

where $\xi$ represents the task embedding extractor. For each sample $(X^i, Y^i)$ in the support set $\mathcal{S}_{\mathcal{T}}$ from any task $\mathcal{T}$, $\xi$ first transforms $Y^i \in \mathbb{R}^{H \times W \times C_{\mathcal{T}}}$ into $\widetilde{Y}^i \in \mathbb{R}^{H \times W \times 3}$ using linear projections. The specific details of this transformation are provided in the appendix. Then, $X^i$ and $\widetilde{Y}^i$ are concatenated, resulting in an input with shape $\mathbb{R}^{H \times 2W \times 3}$. Next, this concatenated input is fed into a vision transformer [5] consisting of three stacked transformer blocks. An extra token is appended to the sequence of embedded patches, whose state at the output of the transformer serves as the task token. For each sample in the support set $\mathcal{S}_{\mathcal{T}}$, a task token is extracted. Finally, $\xi$ averages all these task tokens to output the final task embedding.

The inclusion of task-conditioning allows the meta-routers to dynamically adapt themselves using task embeddings, thereby increasing their flexibility. This enhancement leads to improved adaptation performance compared to the naive meta-routers, which rely solely on fine-tuning a fixed set of initialization weights. Note that our method differs from existing task-conditioned routers [8, 50] in a fundamental way. These existing methods extract the task embedding in a dictionary-like manner: $embed(id_{task}(x))$, where $id_{task}(x)$ is the task index of current token $x$ and $embed(\cdot)$ represents the embedding layer. However, in the few-shot learning setting, the pre-trained embedding layer $embed(\cdot)$ is not adaptable to novel tasks. In contrast, our approach directly extract the task embedding from the support samples themselves, making it well-suited for few-shot scenarios.

During meta-training, we only optimize the parameters of the meta-routers, the task embedding extractor and the task-specific convolution layers in the prediction head. Note that the task embedding extractor remains fixed during the fine-tuning procedure $\hat{\mathcal{F}}(X_Q; \mathcal{S}_{\hat{\mathcal{T}}})$ and the testing phase. This ensures a consistent task embedding space, enabling meta-routers to achieve more efficient adaptation when prompted with a stable task embedding.

**Router fine-tuning.** In the testing phase, given an arbitrary unseen dense prediction task $\tilde{\mathcal{T}}$ with $N_S$ support examples, the model needs to rapidly adapt to this new task while avoid overfitting. Thanks to the versatile backbone acquired through multi-task pre-training and the meta-routers trained with episodes-based meta-learning, adapting the model to unseen few-shot tasks are super easy. You just need to fine-tune the meta-routers and the task-specific convolution layer on $N_S$ support samples of the novel task $\tilde{\mathcal{T}}$ for $N_{FT}$ steps, which is consistent with the setup in the meta-training stage.

# 5 EXPERIMENTS

## 5.1 Experimental Setup

**Datasets.** We evaluate our approach on a variant of the Taskonomy dataset [45], which only uses the Taskonomy-tiny partition. Taskonomy dataset contains indoor images with various annotations, where we choose ten dense prediction tasks of diverse output dimensions: semantic segmentation (SS), euclidean distance (ED), Z-buffer depth (ZD), texture edge (TE), occlusion edge (OE), 2D

**Table 1: Quantitative comparison on a variant of Taskonomy dataset. 10-shot results are evaluated on each fold after being trained on the tasks from the other folds, while fully-supervised methods are trained on tasks from each fold independently (DPT) or trained on all folds (InvPT). The bolded results are the best results within their respective supervision settings.**

| Supervision | Method | Tasks | | | | | | | | | |
| --- | --- | --- | --- | --- | --- | --- | --- | --- | --- | --- | --- |
| | | Fold 1 | | Fold 2 | | Fold 3 | | Fold 4 | | Fold 5 | |
| | | SS | SN | ED | ZD | TE | OE | K2 | K3 | RS | PC |
| | | mIoU↑ | mErr↓ | RMSE↓ | RMSE↓ | RMSE↓ | RMSE↓ | RMSE↓ | RMSE↓ | RMSE↓ | RMSE↓ |
| Full | DPT [27] | **0.4449** | **6.4414** | **0.0534** | **0.0268** | **0.0188** | **0.0689** | **0.0358** | **0.0357** | **0.0860** | **0.0347** |
| | InvPT [43] | 0.3900 | 12.9249 | 0.0589 | 0.0298 | 0.0517 | 0.0788 | 0.0456 | 0.0384 | 0.0949 | 0.0370 |
| 10-Shot | HSNet [23] | 0.1069 | 24.9120 | 0.2375 | 0.0748 | 0.1746 | 0.1643 | 0.1056 | 0.0651 | 0.2627 | 0.0610 |
| | VAT [14] | 0.0353 | 25.8134 | 0.2718 | 0.0779 | 0.1719 | 0.1655 | 0.1450 | 0.0678 | 0.2709 | 0.0796 |
| | DGPNet [16] | 0.0261 | 29.1668 | 0.4579 | 0.2846 | 0.1881 | 0.2130 | 0.1104 | 0.1308 | 0.3680 | 0.3574 |
| | VTM [18] | 0.4097 | 11.4391 | 0.0741 | 0.0316 | 0.0791 | 0.0912 | 0.0639 | 0.0519 | 0.1089 | 0.0420 |
| | UniDense (ours) | **0.4310** | **9.1261** | **0.0687** | **0.0299** | **0.0634** | **0.0871** | **0.0617** | **0.0442** | **0.1015** | **0.0386** |

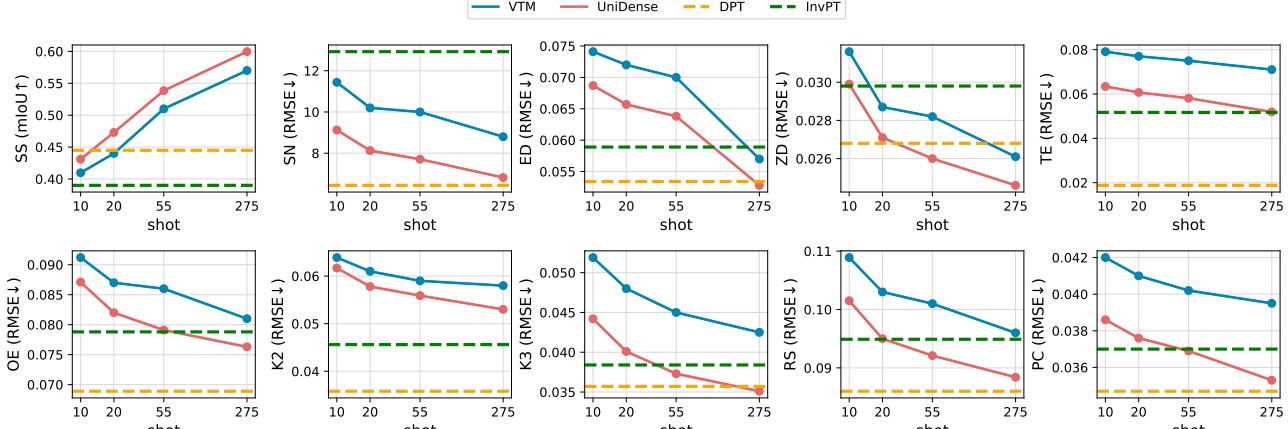

**Figure 3: Performance comparison on various shots. UniDense consistently outperforms the previous SOTA method VTM by a significant margin and even surpasses the fully supervised method InvPT on 9 out of 10 tasks.**

keypoints (K2), 3D keypoints (K3), reshading (RS), principal curvature (PC), surface normal (SN). Following VTM, we choose some single-channel tasks (ED, TE, OE) and transform them into multi-channel tasks to increase task diversity, and standardize all labels to [0, 1]. All images and labels are resized to 256 × 256 resolution. Additional details are given in the appendix.

**Evaluation protocol.** Following the standard evaluation setup of universal few-shot dense prediction [18], we partition the ten tasks to construct a 5-fold split, in each of which two tasks are used for few-shot evaluation and the remaining eight are used for training. The split is shown in Table 1. For evaluation on semantic segmentation (SS), we follow the standard binary segmentation protocol in few-shot semantic segmentation [32] and report the mean intersection over union (mIoU) for all classes. We use the mean angle error (mErr) for surface normal prediction (SN) [6], and use root mean square error (RMSE) for other tasks. Note that all metrics are better when their values are smaller except for the metric of SS. To be consistent with VTM [18], we test our method on the 'muleshoe' building of Taskonomy dataset.

**Implementation details.** We use the released 1-5 version of Stable Diffusion in our experiments. We introduce MoE attention layers and MoE MLP layers into $\epsilon_\theta$ every two layers. For MoE attention layers, we use 24 experts with top-k as 8. For MoE MLP layers, we use 12 experts with top-k as 4. For the multi-task pre-training,

we set the learning rate to 1e-4 with a batch size of 16. For each training batch, we randomly sample 16 images with their labels for all training tasks. We use 5 warmup epochs with 50 total training epochs. For the meta-training of meta-routers, the learning rate is set as 5e-4, with 3000 episodes in total. Recent research [21] has found that utilizing a small learning rate exclusively for the non-finetuned portion of the model can result in significant performance enhancement during the few-shot fine-tuning process. Therefore, we employ a 1/100 smaller learning rate to the rest part of the model for model fine-tuning in the testing phase. The fine-tuning steps $N_{FT}$ is set as 30. Further details can be found in the appendix.

## 5.2 Comparison with State-of-the-Art Methods

In Table 1, we compare the 10-shot performance of our model with current State-Of-The-Art (SOTA) methods. Among them, 'HSNet', 'DGPNet' and 'VAT' are adapted from previous SOTA few-shot segmentation methods, following [18]. Our UniDense method consistently outperforms the previous SOTA few-shot method VTM across all tasks. Figure 3 illustrates the performance of UniDense and current SOTA methods as we increase the size of the support set from 10 to 275. Our method exhibits a significant advantage over VTM across all shot numbers. Remarkably, our model even surpasses the fully supervised method InvPT on 9 out of 10 tasks using a much smaller dataset (0.1%). This implies the potential value

**Table 2: Comparing different few-shot adaptation methods while controlling all models' backbone as ViT [5].**

| Method | Backbone | SS (mIoU↑) | SN (mErr↓) |
|---|---|---|---|
| VTM w/o FT [18] | ViT-B | 0.0002 | 23.4212 |
| Painter [40] | ViT-L | 0.1167 | 19.0734 |
| Linear Head | ViT-B | 0.2681 | 13.0704 |
| VTM [18] | ViT-B | 0.4097 | 11.4391 |
| UniDense-ViT | ViT-B | **0.4158** | **10.2803** |

**Table 3: Effectiveness of meta-routers.**

| Method | SS (mIoU↑) | SN (mErr↓) |
|---|---|---|
| Random Router | 0.3946 | 10.5673 |
| Meta-Router w/o Cond. | 0.4239 | 9.3122 |
| Meta-Router | **0.4310** | **9.1261** |

of our method in specialized domains (e.g., medical images) where the number of available labels ranges from dozens to hundreds.

## 5.3 Ablation Study

Our method's superior performance is attributed to the efficient few-shot adaptation method of router fine-tuning with meta-routers and the reliable backbone based on the Stable Diffusion model. This section conducts ablation experiments to investigate the individual effectiveness of each design.

**Comparing adaptation methods with same backbone.** Due to the architectural differences between our method and VTM, a direct comparison may be unfair. To demonstrate the effectiveness of our few-shot adaptation strategy, we perform an ablation experiment by replacing the stable diffusion backbone with a ViT backbone. Specifically, we customize MoE attention layers and MoE MLP layers within the transformer blocks of the DPT-Base model [27] to create a MoE ViT model. We then apply our three-stage UniDense pipeline to this MoE ViT model, resulting in the 'UniDense-ViT' model. Table 2 presents the results. The upper section of methods represents few-shot adaptation without fine-tuning, where Painter [40] adopts the in-context learning paradigm. The lower section of methods involves fine-tuning for few-shot adaptation, where 'Linear Head' denotes the method that fine-tunes only the linear mapping layer while freezing the backbone. Our UniDense-ViT method outperforms VTM with the same ViT-B backbone and achieves a significant improvement over the advanced in-context learning method Painter. These results validate the effectiveness of our few-shot adaptation strategy.

**Effect of Meta-Routers.** To realize the idea of few-shot adaptation by selecting task-specific computation pathways, we propose to fine-tune the routers of the MoE backbone. However, this raises questions about which router to fine-tune and how to ensure its effectiveness. Here, we compare three different ways to construct routers for few-shot fine-tuning in Table 3. 'Random Router' represents randomly initializing a set of routers for few-shot fine-tuning. 'Meta-Router w/o Cond.' represents using meta-routers without task conditions. As observed in the table, meta-routers significantly surpass randomly initialized routers, and the involvement of task conditions further improves the performance.

**Comparing different methods exploiting Stable Diffusion.** While the pre-trained knowledge of Stable Diffusion (SD) greatly facilitates our method, it is worth exploring if there are more

**Table 4: Comparing different methods to leverage SD pre-trained knowledge.**

| Method | SS (mIoU↑) | SN (mErr↓) |
|---|---|---|
| SD+FT | 0.3869 | 10.8713 |
| SD+MTL+FT | 0.4016 | 10.2134 |
| UniDense | **0.4310** | **9.1261** |

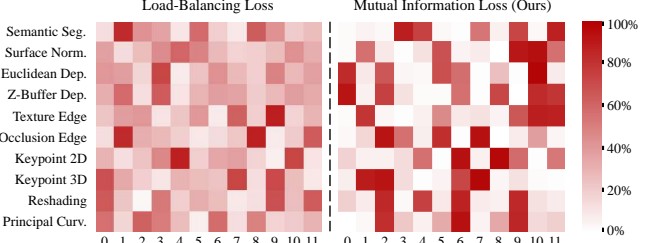

**Figure 4: Visualization of the frequency that experts being selected for each few-shot task. The horizontal axis represents the index of the expert in a MoE MLP layer.**

direct and effective ways to leverage the SD pre-trained knowledge. Here, we compare three different methods leveraging SD pre-trained knowledge in universal few-shot dense prediction. In Table 4, 'SD+FT' involves directly fine-tuning the Stable Diffusion model (specifically, fine-tuning the UNet and the last convolution layer in decoder) for few-shot adaptation, without multi-task pre-training. 'SD+MTL+FT' first performs multi-task pre-training on the SD backbone and then fine-tunes it for novel few-shot tasks. Note that these two methods do not use the MoE architecture. As observed in the table, UniDense significantly outperforms the other methods, demonstrating that our UniDense framework fully exploit SD pre-trained knowledge and effectively adapt the SD model into the task of universal few-shot dense prediction.

## 5.4 Visualization

In this section, we perform visualization experiments to demonstrate that our method is capable of extracting task-specific knowledge by selecting relevant experts for different tasks, and that our method can effectively transfer knowledge between relevant tasks. We also visualize the predictions generated by our method to demonstrate its effectiveness.

**Activation frequencies of experts for different tasks.** One advantage of our method is the ability to extract task-specific knowledge by selecting relevant experts for different tasks. To demonstrate this, we visualize the frequencies of activated experts for different few-shot tasks. In Figure 4, the left part displays the expert frequencies obtained using the load-balancing loss, which promotes similar utilization frequencies of experts across different tasks. In contrast, the mutual information loss (right) adopted by our method encourages expert specialization, resulting in a sparse but strong dependence between experts and tasks.

**Transfer efficiency of expert knowledge.** We are interested in understanding which task's knowledge is more useful than others for a specific task. To address this, we define the transfer efficiency of expert knowledge as the percentage of experts that the target few-shot task shares with the pre-training task. Figure 5 presents the results for all data folds. From the table, we observe that there is

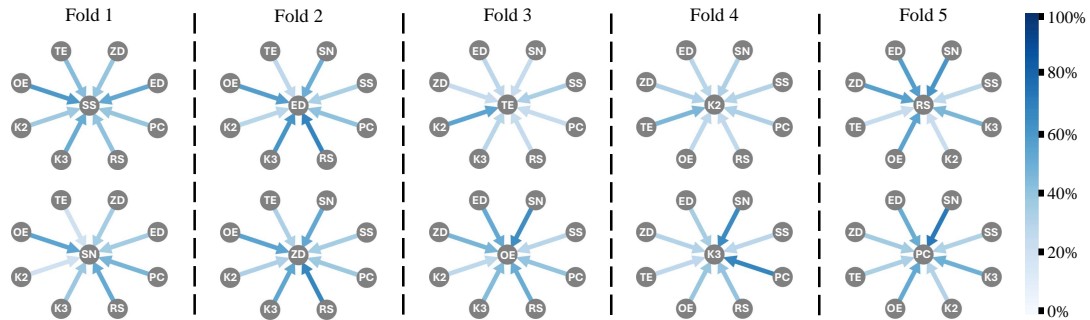

Figure 5: Visualization of the transfer efficiency of expert knowledge between different tasks. The tasks at the tail of the arrow are the tasks used for multi-task pre-training, while the tasks at the head of the arrow are novel few-shot tasks.

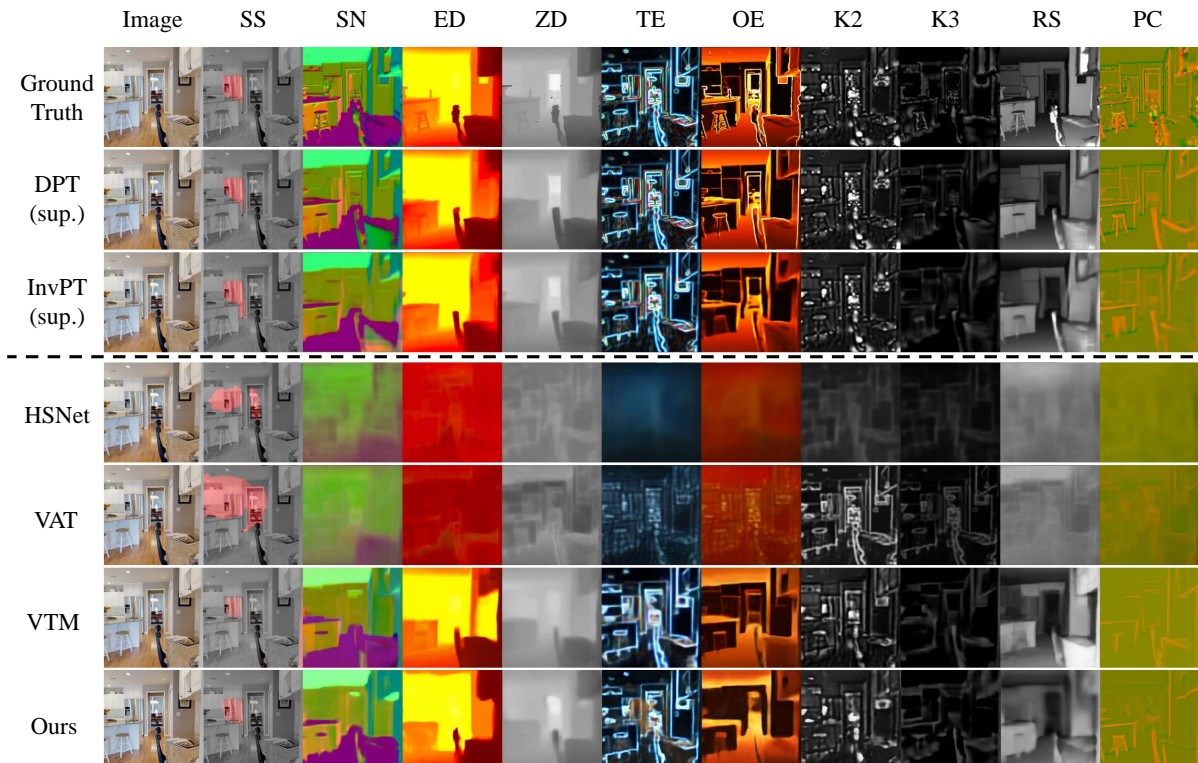

Figure 6: Qualitative comparison of few-shot learning methods in the 10-shot setting.

a higher transfer efficiency between 2D tasks (TE, K2) and between 3D tasks (ED, ZD, SN, RS, PC, K3, OE). This finding is reasonable and demonstrates that our framework can effectively transfer useful knowledge between relevant tasks.

**Qualitative comparison.** In Figure 6, we present visualizations of the predictions generated by our method and previous SOTA methods. Comparing our method to VTM, we observe that our method produces more accurate predictions across all tasks, which aligns with the results reported in Table 1.

## 6 CONCLUSION AND DISCUSSION

In this paper, we tackle the universal few-shot learning problem from a novel perspective. Unlike traditional few-shot fine-tuning

techniques that directly use all parameters of the pre-trained model and modify a specific set of weights, our approach focuses on selecting task-relevant computation pathways while keeping the model weights frozen. To achieve this, we introduce a novel framework called UniDense. Our method utilizes the Stable Diffusion model to construct a versatile MoE architecture and fine-tunes its routers to achieve few-shot adaptation. To ensure efficient router fine-tuning, we leverage episodes-based meta-learning to train a set of routers known as Meta-Routers. Additionally, we introduce a task embedding extractor that generates task conditions directly from support samples for the meta-routers. By fine-tuning these meta-routers for novel tasks, we enable highly efficient adaptation of the entire model. Experimental results on a modified version of the Taskonomy dataset demonstrate the superiority of our method.

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
