# OpenReview forum: "UniDense: Unleashing Diffusion Models with Meta-Routers for Universal Few-Shot Dense Prediction"
_acmmm.org/ACMMM/2024/Conference — MM2024 Poster_

### Official Review · Reviewer_okj8 · 2024-05-17

**Rating:** 4
**Confidence:** 2

**Summary:**

This paper introduces a UniDense framework for the universal few-shot dense prediction problem.  The UniDense framework utilizes a versatile MoE architecture based on the Stable Diffusion model to select the task-specific computation pathway for few-shot adaptation. It employs episodes-based meta-learning to train Meta-Routers for this selection and uses task embeddings to dynamically adjust the routers. Experimental results across 10 different dense prediction tasks validate the effectiveness of UniDense.

**Strengths:**

Strengths
1.	Introducing a pre-trained Stable Diffusion model to address the few-shot dense prediction problem is particularly compelling, especially with the incorporation of MoE attention mechanisms and MLP layer transformations within the Stable Diffusion architecture.
2.	Extensive experiments show that the proposed method outperforms other baselines.

**Limitations:**

Weaknesses
1.	In section 4.2, lines 314-317 mention that the model does not utilize the noise addition process of the Stable Diffusion model and also omits the language interface. This raises the question of whether it can still be considered a diffusion model. Without the diffusion process and relying solely on the UNet structure, the claim of using a diffusion model for dense prediction seems somewhat ambitious.
2.	How does the task embedding extractor help the meta-routers? What is the difference between Stage B and Stage C in Figure 2 regarding the task embedding extractor's function?
3.	[Minor] It would be better to explain that MoE stands for Mixture of Experts in the abstract. (L23)

**Suitability:**

3

---

### Official Review · Reviewer_MU3x · 2024-05-22

**Rating:** 4
**Confidence:** 3

**Summary:**

UniDense handles few-shot dense prediction tasks using three-stage training and meta-routers. In the first stage of training, balancing weights are updated, while in the second and third stages, meta-routers are refined. Meta-routers initialize routers through meta-learning. UniDense is the first to adopt Stable Diffusion for few-shot dense prediction tasks.

**Strengths:**

- Utilizing task embeddings from the support samples for meta-routers demonstrates a novel approach.
- The performance in the Taskonomy benchmark is impressive, achieving the highest scores across the tasks.
- The overall visualizations and figures are clear and helpful for understanding.

**Limitations:**

- The adoption of Stable Diffusion is mentioned as one of the main contributions. However, stable diffusion is adopted in rather a parallel manner with newly proposed methods, three-stage training and meta-routers. Can the authors explain more about the connection between the newly proposed methods and diffusion models?
- Several architectures or equations have been selected from other works, such as SD, MoA, and mutual information loss from Mod-Squad, making it difficult to assess the performance boost attributed to the main methods. In Table 4, VTM with a ViT backbone and UniDense without MoE may seem to achieve comparable scores. Additionally, scores from other folds (Fold 2-5) would further clarify the ablation study.

**Suitability:**

3

---

### Official Review · Reviewer_8VBY · 2024-05-24

**Rating:** 3
**Confidence:** 2

**Summary:**

Facing the risk of disrupting the pre-trained knowledge, this paper introduces UniDense for universal few-shot dense prediction. UniDense constructs a versatile MoE architecture and utilizes episodes-based meta-learning to train a set of meta-routers for the MoE model. The paper focuses on selecting the task-relevant computation pathways of the pre-trained model while keeping the model weights frozen. First, UniDense constructs a versatile MoE architecture for dense prediction based on the Stable Diffusion model. Then UniDense utilizes episodes-based meta-learning to train a set of routers for this MoE model, called Meta-Routers. Moreover, for each few-shot task, UniDense leverage support samples to extract a task embedding.

**Strengths:**

(1) The paper is logically.

(2) Sufficient experimental analysis.

**Limitations:**

(1) The method is relatively complex, and a set of Meta-Routersfor requires additional resource support.

(2) Except the introduction of diffusion model, other improvements are limited.

**Suitability:**

2

---

### Official Review · Reviewer_NnZt · 2024-05-24

**Rating:** 4
**Confidence:** 2

**Summary:**

The paper introduces "UniDense," a novel framework designed to enhance the performance of few-shot dense prediction tasks across a variety of domains such as semantic segmentation, depth estimation, and surface normal prediction. The core innovation of UniDense lies in its use of diffusion models coupled with a meta-routing mechanism that dynamically adapts the diffusion process based on the task at hand. This approach allows UniDense to efficiently leverage a small number of examples to make accurate dense predictions, showcasing its versatility and effectiveness in a few-shot learning context.

**Strengths:**

1. The introduction of meta-routers to guide diffusion models in the context of dense prediction tasks is an innovation. This approach addresses the challenge in few-shot learning by enabling more effective adaptation to new tasks with minimal examples.
2. The methodologies proposed in the paper are technically sound. The authors detail the implementation of UniDense, ensuring reproducibility and providing a clear understanding of the model's inner workings.
3. UniDense's ability to perform across different dense prediction tasks highlights its potential for wide-ranging applications in areas such as autonomous driving, robotics, and augmented reality, among others.

**Limitations:**

0. **Not related to multimedia** This paper studies the dense prediction problem, which is unimodal. The authors claimed "This paper explores the application of pre-trained knowledge from the multi-model generative foundation model Stable Diffusion to a downstream task known as universal few-shot dense prediction". However, using stable diffusion for a computer vision problem can not be recognized as a contribution to MM processing.
1. **Can UniDense generalize to Unseen Domains**: While UniDense shows impressive performance on the evaluated tasks, the paper does not extensively discuss its ability to generalize to completely unseen domains or tasks that significantly deviate from the ones tested.
2. **Limited Discussion on Failure Cases**: The evaluation primarily focuses on scenarios where UniDense succeeds. A more comprehensive discussion on its limitations or failure cases, especially in extremely challenging few-shot scenarios, would provide a more balanced view of the model's capabilities.
3. **Lack of Discussion on Computational Resources**: The paper lacks a detailed analysis of the computational resources required for training and inference. It would be beneficial to understand the trade-offs between performance and computational efficiency and also compare with the SOTA in this aspect.

Since I am not quite familiar with the dense prediction task, I vote for borderline acceptance with a low confidence score.

**Suitability:**

1

---

### Meta-Review · Area_Chair_bg64 · 2024-07-03

**Recommendation:** Accept (Poster)
**Confidence:** 4

**Metareview:**

After rebuttal, reviewers agree on the acceptance of this paper, mostly positive comments about the introduction of meta routers. The major caveat is that it is a relatively complex method and potentially requires large computational resources. However, the merit of the meta-routing mechanism that dynamically adapts the diffusion process is reasonable to reviewers, moreover after rebuttal the outlook on the paper was more positive. Several ideas were highlighted as interesting hence the AC recommends acceptance.